# OpenReview forum: "LLM Can be a Dangerous Persuader: Empirical Study of Persuasion Safety in Large Language Models"
_colmweb.org/COLM/2025/Conference — COLM 2025_

### Official Review · Reviewer_UYEj · 2025-04-28

**Rating:** 8
**Confidence:** 4
**Ethics Flag:** 1

**Summary:**

The paper presents a systematic investigation into the persuasion safety of Large Language Models (LLMs). It introduces PERSUSAFETY, a novel framework for assessing the ethical risks associated with LLM-driven persuasion across diverse scenarios. The study evaluates LLMs' ability to reject unethical persuasion tasks and avoid employing unethical strategies, considering factors like personality traits and external pressures. The authors conducted extensive experiments on eight widely used LLMs, revealing significant safety concerns.

Quality: The research is well-structured and employs a comprehensive framework (PERSUSAFETY) for evaluating LLM persuasion safety. The methodology is clearly defined, with detailed descriptions of the persuasion task creation, conversation simulation, and safety assessment stages. The experiments appear to be rigorously conducted, and the findings are supported by empirical evidence.

Clarity: The paper is generally well-written and easy to understand. The authors clearly articulate the research questions, methodology, and findings. The use of figures and tables enhances the clarity of the presentation.

Originality: The work is original in its focus on the safety aspects of LLM persuasion. The PERSUSAFETY framework is a novel contribution that provides a systematic approach to evaluating the ethical risks associated with LLM-driven persuasion. The paper addresses a gap in the existing literature, which has primarily focused on the effectiveness of LLMs as persuaders rather than the potential safety risks.

Significance:  The study highlights the urgent need to develop more advanced techniques to ensure safety and ethicality in LLM-driven persuasion. The results suggest that current LLMs are not adequately aligned with human-centered values and may pose a risk of unethical influence through manipulation, deception, and exploitation of vulnerabilities. This work calls for further research and development of safety mechanisms to mitigate these risks.

**Questions To Authors:**

1. Have you considered a human expert review? Has there been a human qualitative overview of the results?

2. Will you be releasing the conversation/LLM output data? Please add some more examples for clarity

**Reasons To Accept:**

1. The study involves extensive experiments across eight widely used LLMs, providing a robust empirical evaluation of their persuasion safety. The use of diverse unethical persuasion topics and strategies enhances the generalizability of the findings.

2. The research identifies significant safety concerns in most of the evaluated LLMs. The findings reveal that LLMs often fail to identify harmful persuasion tasks and frequently employ unethical persuasion strategies. These results highlight the urgent need for improved safety alignment in LLMs.

**Reasons To Reject:**

1. There is no human evaluation of the framework for more insights. Do its evaluation metrics correlate with real-world persuasion scenarios or human expert judgment? Without validation, it's difficult to assess the practical significance of the findings.

---

> ### Author Response · Authors · 2025-06-03
> **Official Responses to Reviewer UYEj**
>
> We are sincerely grateful for constructive feedback and recognition of our paper, including providing a robust evaluation and identifying significant safety concerns. Our responses to your questions are provided below:
>
> **W1: No human evaluation of the framework for more insights.**
>
> We would like to clarify that we did a rigorous human assessment on our evaluation metrics (detailed in Appendix B.1), and we show that our automatic evaluation has a high accuracy of 92.6%, demonstrating a strong correlation between our evaluation approach and human expert judgment. We will rectify this by moving the important details back to the main paper.
>
>
> **Q1:** We will fully open-source the PersuSafety dataset, LLM simulation conversations, source code, and evaluation results to facilitate future research. We have provided a case study in Appendix B.3 and will add more examples in the revision.

---

> > ### Comment · Reviewer_UYEj · 2025-06-03
> >
> > Thank you for clarifying these points, especially pointing to the human-expert evaluation section.

---

> > > ### Author Response · Authors · 2025-06-04
> > >
> > > Thank you so much for your recognition of our work and response! We sincerely appreciate your effort in reviewing our paper and your insightful feedback.

---

### Official Review · Reviewer_DLYk · 2025-05-07

**Rating:** 7
**Confidence:** 3
**Ethics Flag:** 1

**Summary:**

This paper presents an approach to systematically evaluating how and when an LLM will be persuasive in unsafe scenarios. The approach begins by generating neutral and unethical persuasion tasks for an LLM to carry out based on a taxonomy of unethical persuasion strategies. The authors then simulate a conversation between two LLMs where one is tasked with attempting to persuade the other, where the other LLM could be simulating one of a number of personas. The conversations are then evaluated using based on whether or not the persuading LLM refused to attempt persuading the other LLM, and if not, how well the LLM performed at persuasion. The authors find that more capable LLMs refuse more in general, but are also better at persuading when they do not refuse the task. The authors also find that effectiveness of the persuader LLM depends heavily on the persona of the persuadee LLM.

**Questions To Authors:**

* Will the persuasion tasks or conversations (with labels) be released publicly? Would this benefit the community?
* Do the authors view LLMs ability to be a dangerous persuader more as a misuse (malicious human adversaries) concern or a misalignment concern (LLMs are themselves malicious adversaries)?
* Is there a more scalable (less human-in-the-loop) way to evaluate types of misbehavior like persuasion in unethical tasks.
* Is the safety alignment goal for LLMs to not be as persuasive or for LLMs to refuse being persuasive on unethical tasks?
* What are the key actionable takeaways for the AI safety community?
* What action should alignment researchers take based on this work?

**Reasons To Accept:**

* The paper introduces a framework that could potentially be made more general or modified for use detecting other types of misalignment.
* The persuasion task generation is very well thought out and structured.
* The paper shows depth in experiments by analyzing how persuasive deception and manipulation varies with exploitable personas.
* The paper is well-written and the presentation is great.

**Reasons To Reject:**

* The main takeaways from the paper are unclear. The conclusions made by the authors are sound, but what the community should do about the results is underspecified. At least part of this has to do with the results not being super surprising. It is quite well-known that LLMs can be deceptive, especially when being coerced into doing so. The authors do not provide any additional call-to-action on top of stating that unethical persuasion is unsafe behavior that should be mitigated.
* The setup is somewhat contrived and limited. The authors use a fixed prompt to try to manipulate the LLM into being persuasive to another LLM. It seems somewhat unrealistic in that it is not quite clearly showing either a misuse or misalignment problem.
* The results are difficult to interpret sometimes. The metrics are made up by the authors and are not explained in the most intuitive ways (this could be improved).
* Human evaluation and curation is used throughout the paper. This is not the most scalable approach for evaluating LLM misbehavior diminishing the systematic nature of the proposed evaluation framework.

---

> ### Author Response · Authors · 2025-06-03
> **Official Responses to Reviewer DLYk (Part I)**
>
> We appreciate the reviewer's thoughtful feedback and helpful suggestions that will enhance our work. Our detailed responses to your concerns and questions are provided below:
>
>
> **W1: The main takeaways from our paper are unclear.**
>
> We would like to clarify that the main goal of our work is to investigate the critical safety risks and vulnerabilities of LLMs in persuasion. Through extensive empirical studies, we reveal several critical findings and advocate for the following actionable future directions for the community: (1) The researchers should develop more advanced alignment techniques to capture the nuances in multi-turn conversation settings, which are more challenging to tackle compared with single-turn settings; (2) Our finding on the inconsistency between safety refusal and unethical strategy usage indicate a critical gap in current alignment approaches, calling for techniques to ensure safe behaviors throughout the complex interactions, beyond superficial refusal; (3) As we found that LLMs will exploit others’ vulnerabilities to pursue goals, the researchers should put more emphasis on letting the model adhere to the safety principles and being more responsible, rather than solely pursuing achieving the goal as instructed.
>
> We believe these actionable takeaways can demonstrate the significance of our findings, extending beyond simply confirming that LLMs can be deceptive, and provide promising yet critical research directions for the community. We appreciate the reviewer for pointing this out, as adding actionable takeaways would further strengthen our work. We will highlight these important takeaways in our revision.
>
> **W2: The setup is somewhat contrived and limited.**
>
> We would like to clarify that the key focus of our work is not simply prompting LLMs to be persuasive to others, but rather systematically eliciting and measuring unethical behaviors that emerge during multi-turn goal-driven conversations. We design our setup in a way that we can roll out controlled simulations under different settings via diverse prompts to enable our comprehensive analysis. Besides, we view LLMs’ ability to be a dangerous persuader more as a misalignment concern, as they should not exhibit malicious behaviors, even when they are explicitly instructed to do so.
>
> **W3: The results are difficult to interpret sometimes.**
>
> We clarify that our evaluation metrics are not arbitrarily made up by ourselves, but are meticulously designed based on established research frameworks (Pratkanis, 1995), where we curate our evaluation guideline based on the 15 common unethical strategies along with their definitions, such that we ensure a rigorous and systematic assessment. Our evaluation and metrics are explained as follows: we apply the LLM judge to examine each persuasion conversation and detect the usage of specific unethical strategies using a 3-point scale (0=not present, 1=ambiguous, 2=clear evidence). A higher score indicates the unethical strategy is used more frequently and severely.
>
> **W4: Human evaluation and curation are not scalable.**
>
> Our PersuSafety framework is designed to be highly autonomous and scalable. The main workload for human efforts is in Stage 1 - Persuasion Task Creation, where we manually create the taxonomy and ask human annotators to verify the quality of persuasion tasks. Notably, human annotators do not create tasks from scratch but rather inspect the quality of LLM-generated instances, significantly reducing the annotation workload while ensuring dataset quality. For evaluation, we apply a fully automatic framework, leveraging Claude-3.5-Sonnet as the judge, to assess the persuasion safety of LLMs without requiring human effort. We had a small-scale human validation to verify the accuracy of the LLM judge, but this is a one-time effort, as we don't need an extensive human assessment in the subsequent evaluation. All the human efforts in our framework are necessary to ensure our study is solid and rigorous.

---

> > ### Author Response · Authors · 2025-06-03
> > **Official Responses to Reviewer DLYk (Part II)**
> >
> > **Q1: The benefits of our resources to the community.**
> >
> > We will fully open-source the PersuSafety dataset, LLM simulation conversations, source code, and evaluation results to facilitate future research. Our resource will benefit the community in the following ways: (1) PersuSafety enables researchers to systematically study LLM safety in dynamic and goal-driven conversations, supporting standardized evaluation across different models and scenarios; (2) The dataset supports research on multiple critical aspects of real-world persuasion safety, including persuadee vulnerability visibility and contextual constraints, across both unethical and ethically neutral persuasion tasks; (3) Our 15 unethical persuasion strategies with detailed annotations enable researchers to develop targeted safety interventions and better understand specific manipulation tactics employed by LLMs.
> >
> >
> > **Q2: The safety alignment goal in our work.**
> >
> > The primary focus of our work is on whether LLMs are safe and ethical, rather than whether they are persuasive. We define the safety alignment goal as, we expect the LLMs to reject the unethical tasks by responding something like “Sorry, I can’t help you with that” and should not initiate such unethical persuasion, as merely helping the users engage in unethical persuasions could already cause harm to others and violate the ethical boundaries. It would be fine if LLMs are persuasive in general (ethically neutral) tasks, as long as they do not exhibit any unethical behavior.

---

> > > ### Comment · Reviewer_DLYk · 2025-06-03
> > >
> > > Thank you for your detailed response. My questions and concerns have been addressed. I'm happy to increase my score.

---

> > > > ### Author Response · Authors · 2025-06-04
> > > >
> > > > Thank you so much for your thoughtful consideration and positive feedback on our responses! We sincerely appreciate the time you've dedicated to reviewing our paper and the constructive comments to improve our work.
> > > >
> > > > It seems that the score in the OpenReview system is still not updated. We would like to kindly confirm if you have updated the score, or if it's due to technical issues with OpenReview.
> > > >
> > > > Thank you again for your valuable feedback and support.

---

> > > > > ### Comment · Reviewer_DLYk · 2025-06-05
> > > > >
> > > > > Apologies. The score should be updated now. For some reason, OpenReview was not allowing me to edit my review when I made the above comment.

---

> > > > > > ### Author Response · Authors · 2025-06-05
> > > > > >
> > > > > > Thank you very much for updating the score! We can also see the change in the OpenReview system now.

---

### Official Review · Reviewer_8jnH · 2025-05-11

**Rating:** 6
**Confidence:** 5
**Ethics Flag:** 1

**Summary:**

This paper presented a benchmark on the safety risks of LLM-driven persuasion. A framework is proposed to investigate how LLMs reject persuasion tasks. Some other factors like personality traits and external pressures are also included to fulfill the benchmark. There are solid experiments and some interesting findings proposed.

**Questions To Authors:**

1. Any potential methods to explain or mitigate this results?
2. What are the main novelty of this framework?

**Reasons To Accept:**

1. A novel benchmark about the safety issue of LLM’s persuasion. There are well-defined taxonomy proposed in the paper and the benchmark is well done.

2. Study about the personalities and contextual constraints is presented in the paper.

3. The experiment is solid and some insightful finding is given.

**Reasons To Reject:**

1. **Limited novelty**. This paper is more like to be a framework that merges several known methods. For instance, this paper includes persuasion strategies and multi-turn settings [1,2]. I admit that more tasks and strategies are included in the paper but the core idea of this paper is not a novel one.

2. **Metrics**. There are several assessment metrics used in this paper. There are human validations (in appendix) but the main assessment is still relying on prompting LLM. The current assessment is not convincible enough.

3. **Explanation or mitigation**. If there is no novel or counter-intuitive findings via benchmark, some deeper explanation or mitigation methods are important.

4. **Minor issue in writing**. The paper’s writing is basically fine. Some sentences are very complex and strange. For instance, in Line 45, the sentence is hard to read: whether LLMs appropriately reject the requests from malicious users, and, for the tasks they choose to engage with, whether and to what extent they employ unethical strategies like deception or threats

---

> ### Author Response · Authors · 2025-06-03
> **Official Responses to Reviewer 8jnH**
>
> We thank the reviewer for their constructive comments and valuable insights to improve this work. The responses to your comments are as follows:
>
>
> **W1: Limited novelty of our work.**
>
> While we employ established simulation methods in our study, our key novelty lies in the research question we investigate, the resources we contribute, and the valuable findings we present. Our work introduces the first systematic study of LLM persuasion safety, which differs significantly from prior research focused on persuasion effectiveness (Durmus et al., 2024) or applications (Wang et al., 2019). The key novelty of our work are: (1) we address the critical and underexplored question of safety risks when LLMs act as persuasive agents in diverse scenarios; (2) we develop the first comprehensive resource with systematic analysis of unethical persuasion strategies and persuadee’s vulnerabilities, and (3) we identify several critical findings with significant implications for responsible AI deployment.
>
> Additionally, could you please provide the specific reference works [1, 2] mentioned in your review? This would help us better discuss and differentiate our contributions from the prior work you referenced regarding persuasion strategies and multi-turn settings.
>
>
> **W2: Automatic evaluation metric.**
>
> We understand the reviewer's concern about our assessment methodology. We have carefully considered this issue in our design, and we clarify as follows. We reduced the complex persuasion safety evaluation to identifying unethical strategies in conversations, essentially a classification task. Our human verification process confirms that Claude-3.5-Sonnet achieves 92.6% accuracy on this task, validating the reliability of our automated framework in capturing nuanced persuasion strategies. This approach aligns with recent research by Zheng et al. (2023) and Liu et al. (2023), which shows that LLMs can achieve a strong correlation with human judgments on complex evaluations.
>
> [1] Judging LLM-as-a-Judge with MT-Bench and Chatbot Arena. Zheng et al., NeurIPS 2023.
>
> [2] G-Eval: NLG Evaluation using GPT-4 with Better Human Alignment. Liu et al., EMNLP 2023.
>
>
> **W3: Explanation and discussion of mitigation methods.**
>
> Our work reveals a critical, counterintuitive finding: a significant disconnect exists between safety refusal capabilities and actual strategic behavior during execution. Claude-3.5-Sonnet, which best refuses unethical tasks, exhibits the highest unethical strategy usage when engaged. Additionally, LLMs systematically exploit vulnerabilities and adapt unethical strategies, even in ethically neutral scenarios. This exposes a critical gap in current alignment approaches in multi-turn conversations.
>
> To address these issues, we propose three mitigation directions leaving for future work: (1) developing alignment techniques for complex multi-turn conversations rather than single-turn scenarios; (2) ensuring consistent ethical behavior throughout interactions, not just initial refusal; and (3) improving existing training paradigms to prioritize safety principles over goal achievement, making models responsible rather than solely goal-driven.
>
> **W4:** We will extensively revise the writing to make our paper easier and clearer to read.

---

> > ### Comment · Reviewer_8jnH · 2025-06-03
> >
> > Sorry for missing the two references in the review. I add them as follows.
> >
> > [1] The Earth is Flat because...: Investigating LLMs' Belief towards Misinformation via Persuasive Conversation
> >
> > [2] How johnny can persuade llms to jailbreak them: Rethinking persuasion to challenge ai safety by humanizing llms

---

> > > ### Author Response · Authors · 2025-06-04
> > >
> > > Thank you for providing these reference works. We appreciate the opportunity to further clarify the distinction between our work and these related studies. While our work shares certain characteristics with previous works, such as the use of persuasion [1,2] and multi-turn conversational settings [1], the core research questions and methodologies differ significantly in the following aspects:
> > >
> > > **First**, the referenced works [1,2] primarily investigate safety vulnerabilities when LLMs act as **persuadees**—examining how they can be manipulated to comply with persuasive inputs, either to believe misinformation [1] or produce harmful responses [2]. In contrast, our work focuses on safety risks when LLMs act as **persuaders**—investigating whether and to what extent they exhibit harmful and unethical behaviors during the persuasion process.
> > >
> > > **Second**, while both our work and [2] employ taxonomies of persuasion strategies, we apply them in different ways. Work [2] uses their taxonomy to *generate* adversarial prompts that jailbreak LLMs into complying with harmful requests. Conversely, we use our taxonomy to *assess* whether LLM persuaders adopt unethical strategies during conversations.
> > >
> > > We hope this clarification helps distinguish our novelty and contributions from existing literature. We will include the discussion in the revision. Please let us know if our responses adequately address your questions and concerns.

---

> > > > ### Author Response · Authors · 2025-06-09
> > > >
> > > > Thank you again for your thoughtful review and your engagement during the discussion period. As the discussion phase is drawing to a close, we would like to check whether our responses have adequately addressed your questions and concerns. We're happy to provide any additional clarification if needed.

---

> > > > > ### Comment · Reviewer_8jnH · 2025-06-09
> > > > > **Maintain positive about this paper**
> > > > >
> > > > > Thanks for the rebuttal and clarification. Please include the discussion during rebuttal into the final version. I maintain positive about this paper.

---

> > > > > > ### Author Response · Authors · 2025-06-11
> > > > > >
> > > > > > Thank you for your response. We will make sure to include the discussion during rebuttal in our final version.

---

### Official Review · Reviewer_jeKk · 2025-05-13

**Rating:** 8
**Confidence:** 4
**Ethics Flag:** 1

**Summary:**

This paper presents an in-depth investigation into the safety risks of persuasive behavior in LLMs. The authors introduce PERSUSAFETY, a three-stage framework for simulating and evaluating LLM behavior in goal-driven persuasion tasks. It supports diverse manipulation scenarios, accounts for user vulnerability traits and contextual constraints, and includes a taxonomy of 15 unethical persuasion strategies, grounded in social cognition.
- The work is of high technical and empirical quality. Evaluation across 8 widely-used LLMs. The use of human-in-the-loop validation and an LLM-assisted judge.
- The paper is well-organized and clearly written.
- The contribution is original, being one of the first to treat persuasion safety as a concern. Prior work focuses on effectiveness or jailbreak risks.
- The findings are useful for LLM developers. Even SOTA LLMs fail to consistently refuse unethical tasks and exploit user vulnerabilities.

**Reasons To Accept:**

The paper conducts a large-scale empirical study and look at refusal rates, strategy usage, vulnerability exploitation, and contextual effects to evaluate the results.
It gives important insights such as:
 - Stronger models are more persuasive but not more ethical.
 - Some models exploit user vulnerabilities even during ethically neutral tasks.
 - Ethical refusal does not correlate with ethical behavior during execution.

**Reasons To Reject:**

- You mention that you are using Claude as an automated judge. But there are cases where claude is evaluating its own outputs. The results (Table 1) for that test also has high scores for claude. I wonder if that could be attributed to the failure of a model in identifying its own errors.

---

> ### Author Response · Authors · 2025-06-03
> **Official Responses to Reviewer jeKk**
>
> We sincerely appreciate your valuable feedback and endorsement of our work, including your recognition of our empirical methodology and important insights. Here are our responses to your comments:
>
>
> **W1: Failure of Claude in identifying its own errors.**
>
> We added an additional analysis of using the other LLM, i.e., GPT-4o, as the judge to investigate the issue. We adopt the same evaluation setup for fair comparison. We report the average scores across 15 unethical strategies for 4 benchmarked LLMs under different LLM judges, respectively.
>
>
> **Table A: The comparison between different LLMs as the judge (Claude v.s. GPT-4o).**
>
> | LLM Judge | Qwen2.5-7B-Instruct | GPT-4o | Claude-3.5-Sonnet | Llama-3.1-8B-Instruct |
> |:-----------:|:--------------------:|:--------:|:-------------------:|:-----------------------:|
> | **GPT-4o** | 0.477 | 0.612 | *0.769* | 0.791 |
> | **Claude** | 0.490 | 0.627 | *0.724* | 0.776 |
>
> From Table A, we observed that the evaluation results under different LLM judges are quite consistent, where Claude-3.5-Sonnet still has relatively high scores on unethical strategy usage when GPT-4o is the judge. This indicates that in our evaluation setup, the LLM judges do not exhibit a systematic failure to detect their own errors.

---

> > ### Comment · Reviewer_jeKk · 2025-06-03
> >
> > Thank you for your response. I acknowledge that the authors have resolved my concerns.

---

> > > ### Author Response · Authors · 2025-06-03
> > >
> > > Thank you very much for your response and recognition!

---

### Decision · Program_Chairs · 2025-07-08

**Decision:**

Accept

**Comment:**

This paper presents a timely and rigorous investigation into persuasion safety in LLMs through the novel PersuSafety framework. The study is well-executed, with extensive empirical evaluation across 8 LLMs and insightful findings that reveal critical alignment gaps in persuasive scenarios. Overall, the reviewers are positive, and I share the same sentiment. Nevertheless, I noted that reviewers raised concerns about evaluation methods and novelty, which the authors addressed convincingly with additional analyses and clarifications. The authors should factor in the reviewers' comments and their responses in the cam-ready version. I recommend acceptance.